# The Covalent Linking of Organophosphorus Heterocycles to Date Palm Wood-Derived Lignin: Hunting for New Materials with Flame-Retardant Potential

**DOI:** 10.3390/molecules28237885

**Published:** 2023-12-01

**Authors:** Daniel J. Davidson, Aidan P. McKay, David B. Cordes, J. Derek Woollins, Nicholas J. Westwood

**Affiliations:** 1School of Chemistry, University of St Andrews and EaStCHEM, North Haugh, St Andrews KY16 9ST, UK; dd68@st-andrews.ac.uk (D.J.D.); apm31@st-andrews.ac.uk (A.P.M.); jdw3@st-andrews.ac.uk (J.D.W.); 2Biomedical Sciences Research Complex, University of St Andrews, North Haugh, St Andrews KY16 9ST, UK; 3Department of Chemistry, Khalifa University, Abu Dhabi 127788, United Arab Emirates

**Keywords:** organophosphorus, heterocycles, lignin, biomass pretreatment, deacylation, click reaction, flame retardants, X-ray crystallography, NMR analysis

## Abstract

Environmentally acceptable and renewably sourced flame retardants are in demand. Recent studies have shown that the incorporation of the biopolymer lignin into a polymer can improve its ability to form a char layer upon heating to a high temperature. Char layer formation is a central component of flame-retardant activity. The covalent modification of lignin is an established technique that is being applied to the development of potential flame retardants. In this study, four novel modified lignins were prepared, and their char-forming abilities were assessed using thermogravimetric analysis. The lignin was obtained from date palm wood using a butanosolv pretreatment. The removal of the majority of the ester groups from this heavily acylated lignin was achieved via alkaline hydrolysis. The subsequent modification of the lignin involved the incorporation of an azide functional group and copper-catalysed azide–alkyne cycloaddition reactions. These reactions enabled novel organophosphorus heterocycles to be linked to the lignin. Our preliminary results suggest that the modified lignins had improved char-forming activity compared to the controls. ^31^P and HSQC NMR and small-molecule X-ray crystallography were used to analyse the prepared compounds and lignins.

## 1. Introduction

Historically, flame-retardant compounds have been toxic and persistent in the environment, with polyhalogenated/polybrominated flame retardants being a well-documented issue [1]. These compounds are now largely banned or heavily restricted; therefore, replacements are required. Organophosphorus flame retardants (OPFRs) [2,3,4,5] are being developed as less toxic and less harmful alternatives, although this class of compounds is not concern-free [6,7]. The main effect of OPFRs likely occurs in the condensed phase via the degradation of the phosphorus motif and polymerisation of resulting free phosphoric acid-containing units to form a char layer. This layer insulates the flammable substrate from the required oxygen, disrupting the fire triangle. The incorporation of a nitrogen-containing functional group to form a P-N bond can provide additional gas-phase flame-retardant character. The nitrogen-containing gases, formed upon decomposition at elevated temperatures, dilute the oxygen content in the vicinity of the fire and therefore inhibit flame growth [8,9]. Many OPFRs are physically blended as small molecules into polymers to produce flame-retardant materials (for example, the extensive use of DOPO [10]). More recently, rather than just blending, the chemical attachment of a OPFR to the polymer has been demonstrated, either by covalent [11] or reversible dynamic bonds [12].

Lignin is a renewable biopolymer isolated from biomass alongside cellulose and hemicellulose. A wide range of different pretreatments are used to obtain lignin, including organosolv methods [13,14]. We, and others, have focused on the use of butanol as a sustainably sourced organic solvent that delivers high-quality lignin via a butanosolv pretreatment [15,16,17,18,19,20,21]. Recently, we have extended the butanosolv methodology to enable the use of more unusual biomasses, including cocoa pod husks, a by-product from chocolate manufacturing (Figure 1A) [22].

Importantly, in the context of this work, the simple addition of unmodified lignin to a polymer is known to enhance the flame-retardant properties of the polymer. This is proposed to result from the degradation of the lignin, leading to improved char layer formation [23]. Studies have shown that modification by covalently linking OPFRs to the lignin can lead to materials with flame-retardant properties (Figure 1A and others) [22,24,25]. Butanosolv lignin is highly suited to selective covalent modification as it is soluble in most organic solvents enabling the use of standard reaction sequences. For example, butanosolv (and other) lignins have been used as substrates for grafting on small molecules using click chemistry [22,26,27]. Increasingly, researchers are interested in enhancing the inherent flame-retardant properties of lignin through its covalent modification with OPFRs.

The work presented here is dedicated to our excellent colleague at the University of St Andrews, Professor Derek Woollins. Derek’s interests continue to be wide-ranging and include the synthesis of phosphorus-, selenium-, or tellerium-containing heterocycles [28,29,30]. Here, we present the synthesis of novel P-heterocycles and the structural analysis of three of these through the use of small-molecule X-ray crystallography. In addition, as a direct result of a collaboration with Derek, we gained access to a relatively understudied biomass source, date palm wood (Figure 1B). We show that an interesting lignin can be obtained by subjecting date palm wood to butanosolv pretreatment, complementing previous work on this lignin type [31]. Through the use of ^31^P NMR spectroscopy methods, a technique frequently used by Professor Woollins [32,33], this lignin was characterised before and after modification with the novel P-heterocycles. Our preliminary assessment of the flame-retardant potential of the novel lignin–OPFR conjugates will guide future work in this developing research area. We would like to thank Professor Woollins for his scientific inspiration and leadership skills.

## 2. Results and Discussion

### 2.1. Phosphorus-Containing Heterocycle Synthesis

The flame-retardant properties of the DOPO motif **1** (Figure 1) are well known [34,35,36], and we have previously reported that after the attachment of O-propargyl DOPO **2** to lignin, the resulting product demonstrates potential flame-retardant properties (Figure 1A) [22]. Based on previous reports [8], the use of N-propargyl DOPO analogue **3** may enable additional gas-phase cooperative flame-retardant activity in this system.

The synthesis and/or use of **3** has been reported in the context of electrode additives [37] and bioactive compound synthesis [38]; however, a slightly modified approach to **3** was used here to convert DOPO **1** to **3** via **4** (Figure 1). A small-molecule X-ray crystallographic analysis of **3** was carried out (Figure 2).

It has been proposed that dibenzo[d,f][1,3,2]-dioxaphosphepine 6-oxide (BPPO)-derived phosphorus heterocycles should also demonstrate flame-retardant properties [39,40]. Novel compounds O- and N-propargyl BPPO **5** and **6**, respectively, were therefore prepared via **7** (Figure 1). The preliminary testing of the use of O-propargyl BPPO **5** in copper-catalysed alkyne–azide click reactions (CuAAC) identified several issues on both models and lignin (see SI for more detail, Appendix A); therefore, the main focus of this study became the modification of lignin by N-propargyl DOPO **3** and N-propargyl BPPO **6**.

### 2.2. X-ray Crystallography

Crystals of **3**, **5**, and **6** suitable for X-ray analysis were grown from ethanol, dichloromethane, or isopropanol solutions of the respective compounds. The compounds crystallised in the monoclinic *P*2_1_/*c*, orthorhombic *Pca*2_1_, and triclinic *P*$\bar{1}$ space groups, respectively, and contain either one (**3** and **6**) or two (**5**) molecules in the asymmetric units (Figure 2). The two aryl groups in **3** are nearly co-planar with a slight twist of 9.77(14)°, with the six membered oxophosphinine ring forming a slightly distorted hexagon (O2-P1-C8 101.90(17)°). In contrast, the two aryl groups in both **5** and **6** show moderate twists of 46.2(2)°, 44.1(2)°, and 42.61(4)°, respectively, and have slightly puckered dioxophosphepine rings (endo-cyclic O-P-O 104.75(19)°, 104.46(19)°, and 102.49(4)°).

Compounds **3** and **6** form hydrogen bonded chains down [1 0 0] and [0 1 0], respectively, through $C\left(7\right)\left[R_{2}^{2}\left(8\right)R_{4}^{2}\left(14\right)\right]$ motifs composed of both moderate strength NH···O (H···O 1.88(2) and 2.017(14) Å, N···O 2.846(4) and 2.9101(13) Å) and non-classical C^sp^H···O (H···O 2.278(3) and 2.2846(8) Å, C···O 3.208(6) and 3.2049(16) Å) hydrogen bonds (Figure 3 for **3**). When viewed down [1 0 0], the hydrogen bonded chains of **3** form a herringbone arrangement. A combination of weaker CH···O (H···O 2.557(3) and 2.707(3) Å, C···O 3.409(5) and 3.617(6) Å) and π-stacking (C···centroid 3.708(4) Å) interactions leads to the formation of sheets in the (1 0 0) plane. The chains of **6** do not adopt a herringbone arrangement and form sheets in the (0 1 0) plane through weak CH···O interactions (H···O 2.5870(8) Å and 2.8830(8) Å, C···O 3.5218(14) and 3.6209(14) Å).

In the structure of **5**, each molecule forms $C\left(7\right)\left[R_{3}^{3}\left(14\right)\right]$ chains down [1 0 0] through non-classical C^sp^H···O (H···O 2.577(4) and 2.544(4) Å, C···O 3.446(13) and 3.437(12) Å) and weaker CH···O (H···O 2.376(4) and 2.378(4) Å, C···O 3.362(10) and 3.353(10) Å) hydrogen bonds, supported by CH···π interactions (H···centroid 3.001(3) Å, C···centroid 3.821(9) Å) (Figure 4). These chains form sheets in the (1 0 0) plane through weak CH···O (H···O 2.588(4)–2.678(4) Å, C···O 3.216(8)–3.489(7) Å), CH···π (H···centroid 2.888(3) Å, C···centroid 3.712(8) Å), and π···π (centroid···centroid 3.723(2) Å) interactions.

### 2.3. Model Compound Synthesis

Due to the complex heterogeneous structure of lignin [41], the assignment of structural features via NMR analysis is aided by the preparation and NMR analysis of simplified model compounds [42]. We have previously prepared models of the butoxylated β-O-4 linkage modified with various functional groups at the γ-position, including **8**, which contains an azide functionality that can be utilised in copper-catalysed azide–alkyne cycloaddition (CuAAC) click reactions (Figure 1) [43]. Novel model compounds **9**, **10**, and **11** were prepared from **8** and characterised for comparison with the modified lignins (Figure 1).

### 2.4. Lignin Substrate Preparation

Using a procedure previously described in the literature (optimised for unusual biomasses [22]), a sample of date palm wood (DPW) was processed using a butanol pretreatment to prepare a date palm wood lignin (**DPW Lignin**) in good yield (Appendix A). This butanosolv lignin was initially characterised by 2D HSQC NMR and quantitative ^31^P NMR after phosphitylation using a procedure previously described in the literature [44,45] (Figure 5A–C and Appendix A). Whilst the aliphatic OH content of this lignin was reasonably high (6.8 mmol/g, Figure 5A,B), it was observed that many of the potentially modifiable β-O-4 sites were acylated (Figure 5C).

These acyl groups were expected based on previous reports that have shown that date palm lignins contain a range of ester pendant groups at the γ-position of the β-O-4 units, including abundant benzoate and *p*-hydroxybenzoate esters, as well as minor components such as vanillic and syringic esters [31]. Inspired by well-established methods of hydrolysing ester units in lignin [31,46,47], aqueous sodium hydroxide solution was used with a sample of **DPW lignin** to produce a deacylated lignin (**DeAcyl Lignin**). Following this reaction, there was a nearly 40% increase in aliphatic OH content (from 6.8 to 9.4 mmol/g, Figure 5B), with the acylated β-O-4 linkage content (labelled *p*-BH in Figure 5) having decreased. Detailed HSQC and HMBC NMR analyses of the aqueous component after hydrolysis allowed for identification of the free benzoic, *p*-hydroxybenzoic, and syringic acids that were cleaved from the lignin (Appendix A). The identification of the free acids facilitated the assignment of the corresponding ester moieties in the aromatic region of the HSQC NMR spectra of the lignin (Appendix A). Whilst each of these ester moieties have been identified in palm lignins before, the expected relative abundance differed compared to previous reports [22]. No *p*-coumarate or ferulate esters were detected in the HSQC NMR analysis, possibly suggesting that these esters were more facile to hydrolyse and may have been removed earlier in the processing of the biomass, as observed with acetyl groups in a previous work [22].

### 2.5. Lignin Modification

Having obtained the required date palm lignins in sufficient quantities, subsequent modification to incorporate azide functional groups was carried out (Figure 2) based on previously established methods [43]. This culminated in the synthesis of **DPW Lignin N_3_** and **DeAcyl Lignin N_3_**, which were characterised by HSQC NMR and IR at each stage to confirm successful modification (see SI for further details; Appendix A).

The modified lignins containing an azide functional handle at the γ-position were then reacted with OPFRs **3** or **6** under CuAAC click conditions to prepare grafted lignins. These modified lignin samples were precipitated and then further purified via column chromatography on silica gel to give final lignins **DPW-3** (118 wt% yield), **DPW-6** (137 wt%), **DeAcyl-3** (130 wt%), and **DeAcyl-6** (137 wt%). Some challenges were encountered at this stage due to the polar nature of both the OPFRs and the final modified lignin (see below and SI for a more detailed discussion).

### 2.6. OPFR-Grafted Lignin Characterisation

The ^31^P NMR and HSQC NMR analyses of model compound **9** (structure in Figure 1) were compared with the NMR spectra of the OPFR grafted lignins obtained from reaction with **3** to determine if the click reaction had been successful. The broad signal in the ^31^P NMR spectrum corresponding to the final DPW lignin (**DPW-3**) obtained upon the CuAAC reaction of **DPW lignin N_3_** with **3** showed good alignment with the signals for model compound **9** (diastereomeric mixture, Figure 6A). However, a sharp signal corresponding to free **3** that was contaminating **DPW-3** was also observed. In addition, overlay of the HSQC NMR analysis of **9** with the final deacylated lignin (**DeAcyl-3**) obtained upon the CuAAC reaction of **DeAcyl lignin N_3_** with **3** also supported a successful reaction (Figure 6B). For example, a signal at ^1^H 4.30-3.85/^13^C 37.2-34.1 corresponded to the methylene hydrogens adjacent to the newly formed triazole ring (Figure 6B). This shows perfect overlay with the analogous signal in **9**. However, the presence of unreacted OPFR **3** was also observed in the HSQC NMR spectra (Appendix A). Analogous results were obtained for the other possible combinations of the lignin azides with the OPFRs (for **DeAcyl lignin N_3_** and **6**, see Figure 6C, and for all other combinations, see Appendix A). Whilst it was gratifying that the CuAAC reaction was successful for all combinations tested, it was disappointing that, despite purifying the final lignins via column chromatography, it was not possible to remove all of the starting small-molecule OPFRs. This observation was in contrast to a previous report on how one can successfully purify OPFR-grafted lignins when using OPFR **2** (Figure 1A and Figure 1) and cocoa pod husk lignin [22]. Presumably, the incorporation of the NHR motif into the OPFR structures (in **3** and **6** compared to **2**) meant that the OPFRs co-eluted with the lignin during purification. Attempts to solve this problem will be the subject of future work.

### 2.7. Thermogravimetric Analysis of OPFR-Grafted Lignins

Despite the presence of small molecular impurities in the final samples of the OPFR-grafted lignins, it was decided to complete this study by carrying out a thermogravimetric analysis (TGA) of the four final lignins (**DPW-3**, **DPW-6**, **DeAcyl-3,** and **DeAcyl-6**). It was proposed that a control TGA experiment would also be carried out, in which a physical mixture (blend) of non-modified **DPW Lignin** and the model compound **12** (Figure 1) would be used. Small molecule **12** represents a compound in which the triazole ring formed in a CuAAc reaction is present, hence enabling the impact of the triazole ring to also be controlled for. This mixture is referred to as the **Control Mixture** below. It was decided that a 5:1 *w*/*w* ratio of **DPW Lignin**/**12** should be used as it was felt that this corresponded to a higher level of small molecule contaminant **12** compared to the amounts of small-molecule OPFRs likely present in the final lignin samples. Any difference in the TGA results (Figure 7) of the final lignins from this **Control Mixture** must be due to the presence of the grafted OPFRs.

A key factor in assessing the potential of a material for use in flame-retardant applications is char formation. This is determined by comparing the mass of sample remaining (as char) after heating the sample to temperatures nearing 1000 °C against suitable controls. Here, three control samples were used: (i) the starting date palm wood lignin (**DPW lignin**), (ii) the starting deacylated DPW lignin (**DeAcyl lignin**), and (iii) the **Control Mixture** discussed above. These controls were compared to the four test lignins: **DPW-3**, **DPW-6**, **DeAcyl-3**, and **DeAcyl-6**. In brief, the two starting lignins (**DPW** and **DeAcyl lignins**) did lead to some char formation, as expected, but this was lower than the amount of char formed by the test lignins. Interestingly, the two best performing lignins were **DeAcyl-3** and **DeAcyl-6**, which are believed to contain a greater amount of OPFRs covalently bonded to the lignin compared to **DPW-3** and **DPW-6**. In addition, both **DeAcyl-3** and **DeAcyl-6** formed an increased amount of char compared to the **Control Mixture**, suggesting that the covalent attachment of the OPFRs to the lignin may provide an advantage over just simply physically mixing OPFRs with lignin. While further work is clearly required to assess the full potential of these materials, one possible explanation for the observed differences is that by holding the OPFR motif closer to the lignin through the use of a covalent bond, the initial degradation reaction is more likely to lead to the intertwining of the lignin- and OPFR-derived chars. This may provide additional structure to the forming char, ultimately improving its formation and therefore the overall char-forming ability of the bulk material.

## 3. Materials and Methods

For a detailed discussion of the lignin experimental procedures, lignin model compounds synthesis, and general experimental considerations, see the Appendix A.

### 3.1. 6-(prop-2-yn-1-ylamino)dibenzo[c,e][1,2]oxaphosphinine 6-oxide 3



DOPO **1** (2.02 g, 9.33 mmol, 1.00 eq.) was dissolved in DCM (25 mL) and cooled to 0 °C under a N_2_ atmosphere. N-chlorosuccinimide (1.37 g, 10.3 mmol, 1.10 eq.) was added slowly portionwise over 10 min, and the resulting mixture was warmed to rt and stirred under N_2_ for 16 h. The resulting suspension was filtered, and the solvent was removed under reduced pressure to afford intermediate **4**, which was used immediately in the next step. Crude **4** was dissolved in fresh DCM (25 mL) and cooled to 0 °C under N_2_, and propargylamine (1.56 mL, 11.2 mmol, 1.20 eq.) and NEt_3_ (0.72 mL, 11.2 mmol, 1.20 eq.) were added slowly dropwise over 10 min then warmed to rt and stirred for 16 h under N_2_. The resulting suspension was filtered, and the filtrate diluted with aq. sat. NaHCO_3_ (20 mL) and extracted with DCM (3 × 15 mL). The combined organic extracts were washed with brine (20 mL) and dried over anhydrous MgSO_4_, and the solvent was removed under reduced pressure. The crude product was purified via column chromatography on silica gel eluting with EtOAc/hexane (0–95%) to afford 6-(prop-2-yn-1-ylamino)dibenzo[c,e][1,2]oxaphosphinine 6-oxide 3 (1.82 g, 72%) as a yellow solid. ^1^H NMR (500 MHz, DMSO-d6) δ 3.18 (1H, t, *J* = 2.5 Hz, H16), 3.67–3.74 (2H, m, H14), 6.24 (1H, dt, *J* = 13.3, 6.8 Hz, H13), 7.27 (1H, dd, *J* = 8.1, 1.3 Hz, H4), 7.29–7.34 (1H, m, H2), 7.42–7.48 (1H, m, H3), 7.56–7.62 (1H, m, H10), 7.74–7.80 (1H, m, H11), 7.82–7.88 (1H, m, H9), 8.16–8.23 (2H, m, H1/12). ^13^C NMR (126 MHz, DMSO-d6) δ 29.59 (C14), 73.55 (C16), 82.30 (d, *J* = 5.4 Hz, C15), 120.23 (d, *J* = 5.9 Hz, C4), 121.97 (d, *J* = 11.5 Hz, C6), 124.12 (d, *J* = 10.7 Hz, C12), 124.42 (C2), 125.42 (d, *J* = 162.6 Hz, C8), 125.48 (C8), 128.41 (d, *J* = 14.1 Hz, C10), 129.66 (d, *J* = 10.0 Hz, C9), 130.48 (C3), 132.91 (d, *J* = 2.3 Hz, C11), 135.96 (d, *J* = 6.8 Hz, C7), 149.36 (d, *J* = 7.0 Hz, C5). ^31^P NMR (202 MHz, DMSO-d6) δ 14.39. IR (ATR) 3229, 3167, 2893, 1597, 1477, 1444, 1213, 1168, 922, 752. mp 147–148 °C. HRMS (ESI) calculated for C_15_H_12_O_2_NPNa [M + Na]+ 292.0503; found 292.0495.

### 3.2. 6-(prop-2-yn-1-yloxy)dibenzo[d,f][1,3,2]dioxaphosphepine 6-oxide 5



2,2′-biphenol (2.00 g, 10.7 mmol, 1.00 eq.) was dissolved in dry THF (40 mL) and cooled to 0 °C under N_2_. POCl_3_ (1.00 mL, 10.7 mmol, 1.00 eq.) was added, followed by the dropwise addition of NEt_3_ (3.00 mL, 21.5 mmol, 2.00 eq.), and then warmed to rt and stirred under N_2_ for 3 h. The resulting suspension was filtered, and the solvent was removed under reduced pressure to afford intermediate **7**, which was used immediately in the next step. Crude **7** was dissolved in fresh dry THF (25 mL) and cooled to 0 °C under N_2_. Propargyl alcohol (0.69 mL, 11.9 mmol, 1.10 eq.) and NEt_3_ (1.64 mL, 11.77 mmol, 1.1 eq.) were dissolved in dry THF (5 mL), and the amine solution was added slowly dropwise over 10 min, then warmed to rt, and subsequently stirred for 16 h. The resulting suspension was filtered, and the solvent was removed from the filtrate under reduced pressure, and the crude product was purified via column chromatography on silica gel eluting with EtOAc/hexane (0–75%) to afford 6-(prop-2-yn-1-yloxy)dibenzo[d,f][1,3,2]dioxaphosphepine 6-oxide 5 (2.26 g, 74%) as an off-white solid. ^1^H NMR (500 MHz, DMSO-*d*_6_) δ 3.90 (1H, t, *J* = 2.4 Hz, H9), 5.02 (2H, dd, *J* = 11.4, 2.5 Hz, H7), 7.40–7.43 (2H, m, H6), 7.47–7.51 (2H, m, H4), 7.58 (2H, dddd, *J* = 8.2, 7.4, 1.7, 0.9 Hz, H5), 7.71 (2H, dd, *J* = 7.7, 1.7 Hz, H3). ^13^C NMR (126 MHz, DMSO-*d*_6_) δ 56.75 (d, *J* = 4.3 Hz, C7), 77.65 (d, *J* = 6.6 Hz, C8), 80.12 (C9), 121.31 (d, *J* = 4.4 Hz, C6), 127.10 (C4), 127.45 (C2), 130.31 (C11), 130.66 (C5), 146.91 (d, *J* = 9.1 Hz, C1). ^31^P NMR (202 MHz, DMSO-*d*_6_) δ 1.82. IR (ATR) 3289, 1476, 1437, 1381, 1292, 1182, 1028, 945, 758. mp 103–105 °C. HRMS (ESI) calculated for C_15_H_11_O_4_PNa [M + Na]^+^ 309.0293; found 309.0279.

### 3.3. 6-(prop-2-yn-1-ylamino)dibenzo[d,f][1,3,2]dioxaphosphepine 6-oxide 6



2,2′-biphenol (2.00 g, 10.7 mmol, 1.00 eq.) was dissolved in dry THF (40 mL) and cooled to 0 °C under N_2_. POCl_3_ (1.00 mL, 10.7 mmol, 1.00 eq.) was added, followed by the dropwise addition of NEt_3_ (3.00 mL, 21.5 mmol, 2.00 eq.), and then warmed to rt and stirred under N_2_ for 3 h. The resulting suspension was filtered, and the solvent was removed from the filtrate under reduced pressure to afford intermediate **7**, which was used immediately in the next step. Crude **7** was dissolved in fresh dry THF (25 mL) and cooled to 0 °C under N_2_. Propargylamine (0.76 mL, 11.9 mmol, 1.10 eq.) and NEt_3_ (1.67 mL, 12.0 mmol, 1.10 eq.) were dissolved in dry THF (5 mL), and the amine solution was added slowly dropwise over 10 min, then warmed to rt, and subsequently stirred for 16 h. The resulting suspension was filtered, and the solvent was removed from the filtrate under reduced pressure, and the crude product was purified via column chromatography on silica gel eluting with EtOAc/hexane (0–80%) to afford 6-(prop-2-yn-1-ylamino)dibenzo[d,f][1,3,2]dioxaphosphepine 6-oxide 6 (2.16 g, 69%) as an orange solid. ^1^H NMR (500 MHz, DMSO-d6) δ 3.28 (1H, t, *J* = 2.5 Hz, H10), 3.69 (2H, ddd, *J* = 14.5, 6.9, 2.5 Hz, H8), 6.47 (1H, dt, *J* = 13.9, 6.9 Hz, H7), 7.32–7.36 (2H, m, H6), 7.41–7.46 (2H, m, H4), 7.54 (2H, dddd, *J* = 8.1, 7.4, 1.7, 0.8 Hz, H5), 7.68 (2H, dd, *J* = 7.7, 1.7 Hz, H3). ^13^C NMR (126 MHz, DMSO-d6) δ 30.21 (C8), 73.77 (C10), 82.11 (d, *J* = 4.4 Hz, C19), 121.77 (d, *J* = 3.7 Hz, C6), 126.38 (C4), 128.04 (C2), 129.91 (C3), 130.22 (C5), 147.45 (d, *J* = 9.3 Hz, C1). ^31^P NMR (202 MHz, DMSO-d6) δ 13.89. IR (ATR) 3248, 3225, 2918, 1477, 1437, 1246, 1184, 999, 918, 758. mp 182–183 °C (decomp.). HRMS (ESI) calculated for C_15_H_12_O_3_NPNa [M + Na]+ 308.0452; found 308.0439.

### 3.4. 6-(((1-benzyl-1H-1,2,3-triazol-4-yl)methyl)amino)dibenzo[d,f][1,3,2]dioxaphosphepine 6-oxide 12



6 (53.8 mg, 0.19 mmol, 1.05 eq.), benzyl azide (23.8 mg, 0.18 mmol, 1.00 eq.), sodium ascorbate (7.70 mg, 0.04 mmol, 0.20 eq.), and copper sulfate pentahydrate (9.40 mg, 0.04 mmol, 0.20 eq.) were dissolved in MeOH (3 mL) and stirred at rt for 12 h. The reaction was diluted with water (10 mL); extraction was carried out with DCM (3 × 5 mL), and the combined organic extracts were washed with aq. sat. NaHCO_3_ (10 mL) and brine (10 mL) before being dried over anhydrous MgSO_4_, and the solvent was removed under reduced pressure. The crude product was purified via column chromatography on silica gel eluting with EtOAc/hexane (0–90%) to afford6-(((1-benzyl-1H-1,2,3-triazol-4-yl)methyl)amino)dibenzo[d,f][1,3,2]dioxaphosphepine 6-oxide 12 (46.8 mg, 63%) as a white solid. ^1^H NMR (500 MHz, DMSO-d6) δ 4.11 (2H, dd, *J* = 13.5, 6.9 Hz, H8), 5.61 (2H, s, H11), 6.46 (1H, dt, *J* = 13.9, 6.9 Hz, H7), 7.13–7.18 (2H, m, H6), 7.31–7.47 (9H, m, H4/5/13/14/15), 7.64 (2H, dd, *J* = 7.5, 1.9 Hz, H3), 8.01 (1H, s, H10). ^13^C NMR (126 MHz, DMSO-d6) δ 36.47 (C8), 52.76 (C11), 121.64 (d, *J* = 3.6 Hz, C6), 122.94 (C10), 126.26 (C4), 128.02 (C13), 128.04 (C2), 128.16 (C15), 128.78 (C14), 129.87 (C3), 130.11 (C5), 136.21 (C12), 146.46 (d, *J* = 4.9 Hz, C9), 147.55 (d, *J* = 9.3 Hz, C1). ^31^P NMR (202 MHz, DMSO-d6) δ 13.99. IR (ATR) 2931, 2870, 1593, 1500, 1437, 1251, 1093, 1024, 935, 785, 754. mp 177–178 °C HRMS (ESI) calculated for C_22_H_19_O_3_N_4_PNa [M + Na]+ 441.1092; found 441.1078.

### 3.5. X-ray Crystallography

X-ray diffraction data for **5** were collected at 173 K using a Rigaku SCXmini CCD diffractometer with a SHINE monochromator [Mo Kα radiation (λ = 0.71073 Å)]. Intensity data were collected using ω steps accumulating area detector images spanning at least a hemisphere of reciprocal space. X-ray diffraction data for **6** were collected at 125 K using a Rigaku FR-X Ultrahigh Brilliance Microfocus RA generator/confocal optics with a XtaLAB P200 diffractometer [Mo Kα radiation (λ = 0.71073 Å)], and data for **3** were collected at 173 K using a Rigaku MM-007HF High Brilliance RA generator/confocal optics with XtaLAB P100 diffractometer [Cu Kα radiation (λ = 1.54187 Å)]. Data for **5** were collected using CrystalClear [48], and for **6** and **3**, data were collected using CrysAlisPro [49]; all data were processed (including correction for Lorentz, polarisation, and absorption) using CrysAlisPro. Structures were solved using dual-space methods (SHELXT) [50] and refined using full-matrix least squares against F^2^ (SHELXL-2019/3) [51]. Non-hydrogen atoms were refined anisotropically, and hydrogen atoms were refined using a riding model, except for the hydrogen atoms on N2 (in both **3** and **6**), which were located from the difference Fourier map and refined isotropically subject to a distance restraint. All calculations were performed using the Olex2 [52] interface. The structure of **5** is in the polar space group *P*ca2_1_ and has an ambiguous flack x parameter (0.13(7)). With the lack of chiral directing groups, the crystal is considered to likely be a racemate. Selected crystallographic data are presented in Appendix A. CCDC 2300932–2300934 contains the supplementary crystallographic data for this paper. These data can be obtained free of charge from The Cambridge Crystallographic Data Centre via the following link: www.ccdc.cam.ac.uk/structures.

## 4. Conclusions

The development of novel flame-retardant materials is important. Here, the potential impact that novel organophosphorus-containing heterocycles bonded to lignin could have in the context of the development of novel flame-retardant materials was assessed. The study began with the synthesis of the phosphorus-containing heterocycles that were analysed using small-molecule X-ray crystallography. The preparation of two different lignins from date palm was then achieved, and both ^31^P and ^1^H-^13^C HSQC NMR methods were used to determine the lignin’s structure. The results of our thermogravimetric analysis revealed that by covalently linking the novel heterocycles to lignin, an increased amount of char was formed compared to lignin alone or a physically mixed control.

## Data Availability

The research data underpinning this publication can be accessed at https://doi.org/10.17630/e0b856df-ea8f-443a-94fc-18bb38436b01.

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
