# Peer review of "The Covalent Linking of Organophosphorus Heterocycles to Date Palm Wood-Derived Lignin: Hunting for New Materials with Flame-Retardant Potential"

_molecules, 2023, doi:10.3390/molecules28237885_

Round 1
Reviewer 1 Report
Comments and Suggestions for Authors
The paper deals with the synthesis of 4 new components from lignin covalent linked with 4 different organophosphorus heterocycles. The subject is original even if authors worked previously on a similar aspect (other type of lignin). The state of art is sufficiently detailed. The part concerning materials and characterization methods is complete. The results are clearly described and well analyzed. The paper is acceptable in state.
Author Response
The paper deals with the synthesis of 4 new components from lignin covalent linked with 4 different organophosphorus heterocycles. The subject is original even if authors worked previously on a similar aspect (other type of lignin). The state of art is sufficiently detailed. The part concerning materials and characterization methods is complete. The results are clearly described and well analyzed. The paper is acceptable in state.
Response We thank reviewer 1 for their comments and consideration of the manuscript, and appreciate the time they have taken to review our work.
Reviewer 2 Report
Comments and Suggestions for Authors
the work is of a good scientific level and may be of interest to Molecules readers.
However, I think the discussion could be improved by considering not only the lignin interunit region in the HSQC spectra, but also the aromatic CH region. The elimination of some acyl moieties (coumaric...) and the addition of aromatic phosphorus moieties could be observed and even quantified (semiquantitative studies). Some information is given in Suppl. Mat, but needs to be better exploited. In particular, in S2 the aromatic CH units (acyl) need to be identified on the basis of data from the literature.
In the discussion and in the formula , Acyl groups = Ac. This is confusing, Ac usually means Acetyl. I think it is better to use "acetyl" and not Ac (example DeAcetyl lignin)
Comments on the Quality of English Language
good
Reviewer 3 Report
Comments and Suggestions for Authors
This article investigates the synthesis of the phosphorus-containing heterocycles, and the potential effect of novel organophosphorus heterocycles combined with lignin on flame retardancy was evaluated. This study is of great significance for the development of new flame retardant materials. I believe it is suitable for publication in this journal. However, there are still some minor issues.
1、The figure 6 should be re-typeset, as it appears to be in draft form.
2、I suggest changing the colors of β-O-4 and β-O-4' in Figure 5 to improve readability.
Author Response
This article investigates the synthesis of the phosphorus-containing heterocycles, and the potential effect of novel organophosphorus heterocycles combined with lignin on flame retardancy was evaluated. This study is of great significance for the development of new flame retardant materials. I believe it is suitable for publication in this journal.
However, there are still some minor issues.
1、The figure 6 should be re-typeset, as it appears to be in draft form.
2、I suggest changing the colors of β-O-4 and β-O-4' in Figure 5 to improve readability.
Response These changes have been made to Figures 5 and 6 and they should now appear more readable on the page.
Round 2
Reviewer 2 Report
Comments and Suggestions for Authors
the manuscript can be published in the present form.